# Managing Devices of a One-to-One Computing Educational Program Using an IoT Infrastructure

**DOI:** 10.3390/s19010070

**Published:** 2018-12-25

**Authors:** Felipe Osimani, Bruno Stecanella, Germán Capdehourat, Lorena Etcheverry, Eduardo Grampín

**Affiliations:** 1Instituto de Computación (INCO), Universidad de la República (UdelaR), Montevideo 11300, Uruguay; felipe.osimani@fing.edu.uy (F.O.); bruno.stecanella@fing.edu.uy (B.S.); lorenae@fing.edu.uy (L.E.); 2Centro Ceibal para el Apoyo a la Educación de la Niñez y la Adolescencia (Plan Ceibal), Montevideo 11500, Uruguay; gcapdehourat@ceibal.edu.uy

**Keywords:** one-to-one computing educational program, Mobile Device Management, Internet of Things

## Abstract

*Plan Ceibal* is the name coined in Uruguay for the local implementation of the One Laptop Per Child (OLPC) initiative. Plan Ceibal distributes laptops and tablets to students and teachers, and also deploys a nationwide wireless network to provide Internet access to these devices, provides video conference facilities, and develops educational applications. Given the scale of the program, management in general, and specifically device management, is a very challenging task. Device maintenance and replacement is a particularly important process; users trigger such kind of replacement processes and usually imply several days without the device. Early detection of fault conditions in the most stressed hardware parts (e.g., batteries) would permit to prompt defensive replacement, contributing to reduce downtime, and improving the user experience. Seeking for better, preventive and scalable device management, in this paper we present a prototype of a Mobile Device Management (MDM) module for Plan Ceibal, developed over an IoT infrastructure, showing the results of a controlled experiment over a sample of the devices. The prototype is deployed over a public IoT infrastructure to speed up the development process, avoiding, in this phase, the need for local infrastructure and maintenance, while enforcing scalability and security requirements. The presented data analysis was implemented off-line and represents a sample of possible metrics which could be used to implement preventive management in a real deployment.

## 1. Introduction

One Laptop Per Child (OLPC) projects involve distributing low-cost laptop computers in less developed countries with the intent to increase opportunities for students. Nicolas Negroponte, the primary advocate for this project, announced his idea of a low-cost laptop at the World Economic Forum in Davos, in 2005, suggesting that laptops and the Internet can compensate for shortcomings in the educational system, and therefore, children should have access to a computer on a daily basis. Following the first XO laptop prototype presented in 2005, there are currently OLPC implementations in 40 countries including Uruguay, Ethiopia, Afghanistan, Argentina, and the United States [1].

Uruguayan government launched Plan Ceibal in 2007 on one single school and rapidly spread over the country reaching full primary schools coverage in the first couple of years. In subsequent years the coverage was extended to high schools, the devices were updated and diversified, comprising several hardware platforms and operating systems. The wireless infrastructure was partially deployed in-house but mainly outsourced to the state-owned telecom operator ANTEL. Managing about one million devices, together with other components of the ICT infrastructure of Plan Ceibal has proven to be a hard task, involving software updates, hardware maintenance and repair, inventory tracking, and many other tasks.

The problem of Information and Communication Technologies (ICT) management is not new. Back in the eighties, as a consequence of the growing dependency of ICT, several guidelines were proposed by governments and the industry, eventually leading to the appearance of the IT Infrastructure Library (ITIL), originated as a collection of books, each covering a specific practice within IT service management [2]. The ICT operations management sub-process enables technical supervision of the ICT infrastructure, being responsible for (i) a stable, secure ICT infrastructure, (ii) an up to date operational documentation library, (iii) a log of all operational events, (iv) the maintenance of operational monitoring and management tools, and (v) operational scripts and procedures. ICT operations management involves many specific sub-processes, such as output management, job scheduling, backup and restore, network monitoring/management, system monitoring/management, database monitoring/management, storage monitoring/management. ITIL processes implementation has been aided by several software suites, which have evolved following the standard frequent updates (https://www.iso.org/committee/5013818/x/catalogue/p/1/u/0/w/0/d/0). The arrival of mobile devices and specifically the Bring Your Own Device (BYOD) movement has presented new challenges to ICT operations management, leading to the appearance of the Mobile Device Management (MDM) concept, and the need of integration with legacy management processes. We will explore these concepts further in Section 2.

The Internet-of-Things (IoT) is a ubiquitous concept that evolves from sensor networks, and includes distributed devices, communications, and cloud platforms for data storage and processing, comprising both analytics and decision-making activities, which may involve actuation over the devices in response to certain conditions [3], fulfilling an *observe-analyze-act* cycle. These concepts are hardly standardized, and the deployment of IoT applications is heavily dependent on verticals, i.e., health, agriculture, smart cities, industries, among others. Mobile devices such as notebooks, tablets, and smart-phones can be considered as sensors with networking and processing capabilities, and therefore may easily accommodate in the IoT architecture; in fact, smart-phones frequently operate as sensor devices in crowd-sourcing applications. Nevertheless, devices usually sense external variables such as temperature, humidity, location, among many others. In the present case, the devices are used to sense internal variables such as CPU usage, battery status, power-on time, and also external variables such as signal strength and WiFi SSIDs in range; push actions, configuration changes, and software updates can also be implemented. Therefore, we focus on a device management module, which can also be integrated into the network management modules, i.e., measuring network traffic. We will further develop these ideas in Section 3 and Section 4.

Our contribution is twofold: on the one hand, we provide an analysis of the applicability of IoT application development to the MDM problem, integrated with a one-to-one computing educational program management. On the other hand, we implement and deploy a prototype solution to the MDM problem using a standard IoT infrastructure. Furthermore, we perform simple, preliminary analytics over the gathered data, which allows envisioning the potential of the proposed solution.

## 2. Management Systems in a One-to-One Program

Managing a one-to-one educational program requires some particular tools and functionalities, besides typical enterprise management systems such as Enterprise Resource Planning (ERP) and Customer Relationship Management (CRM) systems. The current solution at Plan Ceibal considers four key elements: ERP and CRM, devices management, network management, and learning platforms management and integration. Figure 1 summarizes this scenario.

In this section, we first describe the different elements that are currently considered by Plan Ceibal management systems. To give a general understanding of the particularities of this domain, we discuss existent solutions to manage each element, but without the intent to do an exhaustive review. Then, we focus on the new module presented in this work, which enhances the management and monitoring of the devices.

### 2.1. ERP and CRM Systems

ERP systems [4,5] cover the main functional areas of any organization. They are typically composed of several modules that deal with core business processes. For example, the finance and accounting modules handle the budgeting and costing, fixed assets, cash management, billing and payments to suppliers, among others. Another standard module has to do with human resources, involving aspects such as recruiting, training, rostering, payroll, benefits, holidays, days off and retirement. A full ERP system may have many other modules, covering aspects such as work orders, quality control, order processing, supply chain management, warehousing and project management.

CRM systems [6] deal with customer interaction, including aspects such as sales and marketing, and the management of several communication channels (website, phone, email, live chat, social media), which typically involves running a call center or a contact center. Initially, these tasks were part of ERP systems responsibilities. However, given the importance and the complexity of managing customer relationships in almost any business, it is often operated by a different software solution that exchanges data with the ERP.

Plan Ceibal uses standard commercial ERP/CRM solutions. Nevertheless, in the context of this one-to-one program, a fundamental requisite for these systems is the capability to integrate data from the educational system successfully. Students, teachers, and schools act like customers in this scenario, since the main goals of a one-to-one computing initiative are to deliver a laptop or tablet to every teacher and student, and also to provide Internet access to every school. The one-to-one management systems do not own educational system data, but must use it to fulfill its goals, i.e., they must have access to school data (such as location, or prints), and teachers and students data (e.g., listings for each level in every school, and contact info). Due to the sensibility of this information, which includes personal data, data management must take into account the privacy of the program beneficiaries and adhere to existent legislation.

### 2.2. Device Management

The first component specific to a one-to-one program corresponds to the systems that manage delivered devices, which are known as Mobile Device Management (MDM) systems [7,8,9]. Common MDM features include device inventory and monitoring, security functions (e.g., to lock lost or stolen devices), operating system updates, application distribution, and user notifications. This module is particularly critical due to the large number of devices that are usually handled by one-to-one programs. Several commercial MDM solutions are available in the market for major commercial platforms (Android, iOS, Windows, MacOS) installing specific software agents, but do not seem easy to integrate into a general multi-platform environment, and particularly on Linux based systems, which represent the vast majority of the devices delivered by Plan Ceibal. Therefore, no commercial MDM tools are currently deployed; software updates and other simple tasks are performed using home-made tools. For these reasons, it is essential to deploy an MDM module which may help to improve device management, and this is the primary driver for our prototype.

### 2.3. Network Management

The network management module, also specific to this scenario, deals with the administration of the connectivity infrastructure, which provides Internet access, and supports the video-conference equipment and other networking services among schools. The operation and maintenance of all these services typically involve a Network Management System (NMS) [10]. NMS solutions aim to reduce the burden of managing the growing complexity of network infrastructure. They usually collect information from network devices using protocols including, but not limited to, SNMP, ICMP, and CDP. This information is processed and presented to network administrators to help them quickly identify and solve problems such as network faults, performance bottlenecks, and compliance issues. They may also help in provisioning new networks, dealing with tasks such as installing and configuring new equipment and assist in the maintenance of existing networks performing software updates and other tasks.

Typically, network operators and IT administrators deal with several systems and applications to manage their infrastructure, and one-size-fits-all NMS solutions are rare. Most proprietary products have their management systems, which are typically not possible to integrate with other solutions. Moreover, each subsystem such as routing and switching equipment, WLAN solutions and video conference infrastructure has a different management system, and, even for the same provider, all these subsystems are not easy to integrate. Since global management solutions are quite expensive, administrators usually prefer to manage each subsystem separately to avoid this costs. There exist several open-source solutions, and although some of them are very popular, they also face difficulties in integrating with proprietary systems.

### 2.4. Learning Platforms

The fourth component corresponds to learning platforms. In this case, no general solution incorporates the broad range of possibilities involved in providing educational content. On the one hand, we have learning management systems (LMS) which are general content managers for teachers, which enable to create courses and interact with students. On the other hand, intelligent tutoring systems (ITS) are typically tailored for specific topics such as math or language and represent a different platform flavor that should also be managed. In the latter, the goal is to reduce the teacher assistance, automatically guiding the students with previous exercise results. Finally, traditional websites and digital libraries can also be considered educational content providers.

### 2.5. Interactions

While usually each of the previously mentioned systems work independently, it is clear that there must be information exchange between them and data consistency among the data saved in each of the corresponding databases. For example, a router should be present in the ERP database as it corresponds to an asset of the organization, while the same equipment should be identified and monitored in the NMS for its operation. Another clear example is all the information regarding end-users, which is managed by the CRM, but it is also needed for the operation of the learning platforms, with the corresponding privileges for each kind of user. Without breaking this loosely-coupled relationship model, a more advanced integration of these systems would enable better management of the educational program. For instance, it would be nice to integrate the user experiences, typically collected from the CRM, with the network management and operation, thus enabling a Quality of Experience (QoE) [11] based service operation.

## 3. Developing Applications Over IoT Platforms

As already mentioned, the Internet-of-Things (IoT) is a ubiquitous concept that includes (i) distributed devices (a.k.a *things*) that can be identified, controlled, and monitored, and (ii) communications, data storage and processing capabilities that may comprise both analytic and decision-making activities. The IoT is inherently heterogeneous due to the vast variety of devices, communication protocols, APIs, middleware components, and storage options. However, there is also a myriad of approaches to develop and deploy IoT solutions. Ranging from very domain-specific applications (e.g., healthcare domain, traffic, and transportation), to high-level general purpose frameworks and platforms, from open-source solutions to hardware and vendor-specific approaches, the amount of options is enormous.

In this scenario, developers and practitioners must deal with the difficult task of choosing the right tools and approaches to undertake their projects. There are multiple IoT platforms, many of them open-source, which may be deployed and controlled by the application owner. For example, the Hadoop (https://hadoop.apache.org/) ecosystem provides tools that can be combined to fulfill the requirements of an IoT platform. An architecture for smart cities based on these tools is discussed in [12], while in [13] an IoT cloud-based car parking middleware implementation is presented, using Apache Kafka (https://kafka.apache.org/) and Storm (http://storm.apache.org/).

Also, in the last years, the idea of integrating IoT and Cloud Computing has gained momentum [3,14]. Most of the major providers of public Cloud Computing environments started offering IoT features, which try to ease the development and deployment of IoT solutions, exploiting the already available infrastructure. For instance, a cloud-based IoT platform for ambient assisted living using Google Cloud Platform (https://cloud.google.com/) is presented in [15]. We next present an overview of IoT cloud-based platforms, focusing on general-purpose IoT platforms.

### 3.1. An Overview of IoT Cloud Platforms

Despite many standardization efforts, there is still a lack of a reference architecture for IoT applications and platforms. After performing a comprehensive survey on the matter, Al-Fuqaha et al. [16] collect four common architectures. Early approaches applied a simple three-layer architecture, borrowing concepts from network stacks. The authors claim that this approach hinders the complexity of IoT, while the middleware and SOA-based architectures are not suitable for all applications since they impose extra energy and communications requirements to resolve service composition and integration. Finally, they conclude that a five-layer architecture is the most applicable model for IoT applications (Figure 2). We now briefly sketch this approach.

The **Objects** or perception layer represents the physical sensors and actuators that collect data. These objects interact with the **Object Abstraction Layer**, which is responsible for transferring the data produced by the Objects Layer to the Service Management layer through secure channels. Various technologies such as RFID, 3G, GSM, UMTS, WiFi, Bluetooth Low Energy, infrared, or ZigBee are used to transfer data. The **Service Management Layer** pairs services and requesters based on addresses and names, allowing IoT application programmers to work with various objects hiding the specificities of the underlying hardware. Finally, customers and clients either interact with the **Application** or the **Business layers**. The former is the interface by which end-users interact with the devices and query for data (e.g., it may provide temperature measurements) while the latter manages the overall system activities, monitoring and managing the underlying four layers. Usually, this layer supports high-level analysis and decision-making processes. Due to all its responsibilities, the Business Layer usually demands higher computational resources than the other layers.

Since cloud computing environments already provide distributed and scalable hardware and software resources, the idea of implementing some of the layers of an IoT platform using these environments seems straightforward. Cavalcante et al. [17] performed a systematic mapping study on the integration of the IoT and Cloud Computing paradigms, where they focused on two research questions: which are the strategies for integrating IoT and Cloud Computing and which are the existing architectures supporting the construction and execution of cloud-based IoT systems. The authors characterize the integration of IoT and Cloud Computing according to the distribution of responsibilities among the three traditional cloud layers: Infrastructure as a Service (IaaS), Platform as a Service (PaaS) and Software as a Service (SaaS). They identify three integration strategies: minimal, partial and full integration. In the case of the minimal integration, a cloud environment (either in the IaaS or the PaaS) is used to deploy the IoT middleware, and to use it to visualize, compute, analyze, and store the collected data in a scalable way. In the case of partial integration, not only the IoT middleware is deployed in a cloud environment, but the platform also provides new service models based on abstractions of smart objects. Therefore, service models such as Smart Object as a Service (SOaaS) and Sensing as a Service (S2aaS) are provided to hide the heterogeneity of devices and virtualize their capabilities. Finally, the full integration strategy proposes new service models that extend all the conventional Cloud Computing layers to encompass services provided by physical objects, allowing physical devices to expose their functionalities as standardized cloud services. Most of the reviewed approaches either apply the minimal or partial integration strategies. Regarding the architectures that support the construction and execution of cloud-based IoT systems, the authors report that the reviewed solutions are significantly distinct from each other, but the majority of them adopted traditional approaches such as smart gateways, Web services based on REST or SOAP (Simple Object Access Protocol) and drivers or APIs deployed in the SaaS cloud layer. Also, they found that most approaches use the PaaS layer to support the deployment of tools and services for developing applications, as well as the IaaS layer as the underlying infrastructure for hosting and executing applications.

Existing surveys on IoT platforms compare different aspects such as the communication protocols, the device management, and the analytics capabilities, among others; the interested reader may refer for example to [18,19]. In our case, we considered three public cloud IoT platforms, namely Microsoft Azure, IBM Watson, and AWS IoT, and a couple of open-source platforms that can be locally-deployed: Kaa and Fiware. While the later offer a clear advantage concerning privacy and control, they also impose a significant steep learning curve and management efforts for local IT administrators. Given that the primary purpose of this project was to test the feasibility of our approach, we decided to focus on public cloud IoT platforms that allowed us to develop and deploy our prototype quickly.

A detailed analysis of the platforms mentioned above is out of the scope of this paper; nevertheless, among the reasons to choose AWS IoT platform for our prototype, it is worth mentioning the strong device authentication services, the maturity of the documentation, and the usability of Lambda functions. Nevertheless, it is advisable to carefully analyze local deployment of the solution in the commissioning phase of the project, mainly for security and privacy concerns, given that Plan Ceibal manages students and teachers data. In the following, we briefly sketch the main components of AWS IoT platform.

### 3.2. AWS IoT Architecture

AWS IoT is an Amazon Web Services platform that allows to collect and analyze data from internet-connected devices and sensors, feeding that data into AWS cloud applications and storage services [20]. In this approach, most of the layers discussed in Section 3.1 are implemented by developers on the cloud, using the features provided by the platform. We next describe AWS IoT main components and functionalities.

#### 3.2.1. Communications

IoT cloud platforms support multiple communication protocols, usually at least MQTT [21] and HTTP, which permit to send data to the server and directives to the devices. AWS IoT supports HTTP, MQTT and WebSockets communication protocols between connected devices and cloud applications through the Device Gateway, which provides secure two-way communication while limiting latency. The Device Gateway scales automatically, removing the need for an enterprise to provision and manage servers for a pub/sub messaging system, which allows clients to publish and receive messages from one another.

#### 3.2.2. Device Registry

IoT cloud platforms provide a registry of devices, a system of permissions, and mechanisms to add new devices to that registry programmatically. In AWS IoT the Device Registry feature lets a developer register and track devices connected to the service, including metadata for each device such as model numbers and associated certificates. This scheme simplifies the task of adding a new device to the solution, where system administrators only have to provide and install a daemon in the desired devices to perform the registration. The platform will keep track of the devices and their permissions afterward. Also, developers can define a Thing Type to manage similar devices according to shared characteristics.

#### 3.2.3. Authentication/Authorization

AWS requires devices, applications, and users to adhere to authentication policies via X.509 certificates, AWS Identity and Access Management credentials or third-party authentication. AWS encrypts all communication to and from devices. The AWS IoT platform features strong authentication, incorporates fine-grained, policy-based authorization and uses secure communication channels.

#### 3.2.4. Device Shadows

IoT platforms have to deal with the fact that devices may not always be connected. In AWS IoT, Device Shadows provide a uniform interface for all devices, regardless of connectivity limitations, bandwidth, computing ability, or power. This feature enables an application to query data from devices and send commands through REST APIs.

#### 3.2.5. Event-Based Rule Engine

Most IoT cloud platforms provide a rule engine that acts based on events, allowing developers to program behavior into the platform. The capabilities of rule engines vary among platforms, but typical actions include storing in databases, sending messages to devices, and running arbitrary code. In AWS IoT, rules specified using a syntax that’s similar to SQL can be used to transform and organize data. This feature also allows developers to configure how data interacts with other AWS services, such as AWS Lambda, Amazon Kinesis, Amazon Machine Learning, Amazon DynamoDB, and Amazon Elasticsearch Service.

#### 3.2.6. Storage and Analytic Services

Other services, such as databases, can extend IoT platforms capabilities. Together with the rule engine, these services can be used to store data and error logs, process complex events and issue alerts, keep track of firmware versions and update them, and in general to implement the Business Layer features.

#### 3.2.7. Development Environment

AWS IoT developers can manage and develop their solutions with the AWS Management Console, software development kits (SDKs) or the AWS Command Line Interface. The platform also provides AWS IoT APIs to perform service configuration, device registration and logging (in the control plane) and data ingestion in the data plane. Several open-source AWS IoT Device SDKs can be used to optimize memory, power and network bandwidth consumption for devices. Amazon offers AWS IoT Device SDKs for different programming languages, including C and Python.

## 4. Description of Our Proposal

In this section we present our prototype of an MDM module for Plan Ceibal, using AWS IoT infrastructure as the back-end. This module is capable of periodically logging network quality of service metrics (QoS) as seen by the devices, as well as relevant usage metrics, for example, CPU load, RAM usage, battery health, hard drive usage, and application monitoring. Additionally, the module is capable of on-demand data collection: an administrator can order the devices to log at an arbitrary time. A monitoring agent running on the mobile devices, specifically on Linux-based laptops, gathers the data and sends them to a collecting platform. All of the back-end infrastructures reside within the AWS ecosystem, including an MQTT broker, rules over the AWS event-based rule engine, data process and formatting algorithms, logging system, the primary database, and the device discovery and registry modules. The data collecting platform stores the data which may be used to run online or on demand analysis.

The proposed solution meets two fundamental requirements: it can discover managed devices with a standard security level automatically, and it supports to scale up to a million devices. We argue that the features offered by cloud-based IoT platforms, such as Amazon IoT or Microsoft Azure IoT, are well suited to serve as the back-end of our MDM module, mainly because they permit to achieve the scalability requirement easily. Our solution also presents additional functionalities that are not frequent in traditional MDM systems. For example, we can collect network performance data from end-user devices, allowing a significant improvement in the management of the wireless network. Measuring device specific and network specific data with the same tool, potentially avoiding the deployment of additional network monitoring solutions that include specialized and out-of-band sensors like the ones presented in [22,23,24], is a definite advantage. The benefit is twofold. On the one hand, it reduces management budget, and on the other hand, allows us to have a better wireless monitoring solution using measurements from the end-user devices. We next describe the architecture of our prototype.

### 4.1. Prototype Architecture

Figure 3 presents the architecture of our prototype. The **Sensor** layer can be loosely mapped to the **Objects** layer of the five-layer model by Al-Fuqaha et al. [16], while the remaining there layers, namely **Sensor data retrieval**, **Data processing**, and **Data storage** layers permit to implement a **Business** layer on top. In our case, this layer could implement proactive management processes based on online analytics. As mentioned throughout the paper, the prototype implements data retrieval and storage, while we provide some off-line data analysis tasks as a sample of what can be achieved with the system. We now review the architecture.

#### 4.1.1. Sensor Layer

This layer is responsible for data collection using the utilities provided by the Linux OS. We developed a daemon module in *Python* that runs in the computers of Plan Ceibal. It executes periodically using cron, collects data, and sends it to the broker using MQTT messages. The collected data is shown in Table 1. We used standard Linux tools for data collection (including date, ifconfig, iwconfig, acpi, df, and information in /proc, among others) and Eclipse Paho for communication. Regarding portability, changing the back-end would only imply changing the registration and messaging URLs at the device side.

#### 4.1.2. AWS Layers

The three layers implemented over the AWS platform (Sensor data retrieval, Data processing, and Data storage) permit to implement the two primary use cases of the prototype: (i) device registration and (ii) data and error logging. Such architecture, based on platform components, is currently referred to as a serverless architecture. It is worth mentioning that the term serverless has been previously used with different meanings. In particular, within the database community, it refers to database engines that run within the same process as the application, thus avoiding to have a server process listening at a specific port and reducing the communication costs involved in sending requests and receiving responses via TCP/IP or other protocols. SQLite (https://www.sqlite.org), H2 (http://www.h2database.com/), and Realm (https://realm.io/) are examples of this kind of databases. In this project, the term serverless means that the solution does not require the provisioning or managing of any servers by the developers. The servers still exist, but they are abstracted away, and their issues are taken care of by the cloud services provider [25]. This term also encompasses the way server-side logic is executed. In the Function-as-a-Service (FaaS) paradigm, the code is run in event-triggered stateless containers that are managed by cloud service providers. AWS Lambda is Amazon implementation of FaaS. In the following, we describe how we use AWS components to implement the use cases.

##### Devices Registration

The prototype extends AWS Device Registry functionality, developing a *device discovery feature*. Devices register automatically; on install, the daemon registers the device in the platform using its serial number, obtaining credentials and a topic to publish new data. Administrators only need to install the daemon on the devices, and the platform will keep track of them, scaling automatically with the demand. Besides, the devices subscribe to topics, which allows the system to send them messages. Plan Ceibal staff were particularly interested in these features for on-demand data collection and software updating.

A daemon can be implemented, using this approach, for any device with an Internet connection. The only requirements are the proper registration and to send data to the right endpoint. Any device can register, but policies that restrict the access are set on the server side. AWS API Gateway provides a simple and effective way to create and manage *RESTFul* APIs that scale automatically. We developed an API capable of managing HTTP POST requests, registering the devices and creating unique certificates for the devices to connect to the broker. A POST request to the API triggers a *Lambda function* that creates an *X.509 certificate*, used to establish MQTT connections between devices and the broker. It also attaches a policy—a JSON document specifying a device’s permissions in AWS—to that certificate, so the device will only be allowed to publish to a specific MQTT topic created for it. Figure 4a shows a flow diagram for this process.

##### Data and Errors Logging

Our implementation uses AWS IoT MQTT broker and the event-based AWS IoT Rules Engine to process collected data and log errors. First, the Rules Engine sorts incoming messages from already registered and authenticated devices by their type, according to which MQTT topic they are sent. Then, data messages trigger a Lambda function that processes the data, formats it and stores it in a PostgreSQL database running inside AWS RDS, while error messages are forwarded and stored in DynamoDB.

To log errors, messages are sent to device-specific MQTT broker topics that adhere to the following pattern: devices/<CERTIFICATE-ID>/errors. Then, the AWS IoT Rules Engine triggers a custom-made rule that stores the error message in a DynamoDB table. Table 2 shows this rule. Line 2 contains a pseudo-SQL query that describes the messages that this rule should process; in this case all the messages from any errors topic. Lines 5–14 describe the actions to perform. The roleArn parameter identifies a role entity inside AWS that has write access on the errors-table, while lines 9 to 12 specify the key-value pair to store (the device -id and the error message) and the timestamp.

**Listing 1 sensors-19-00070-t002:**
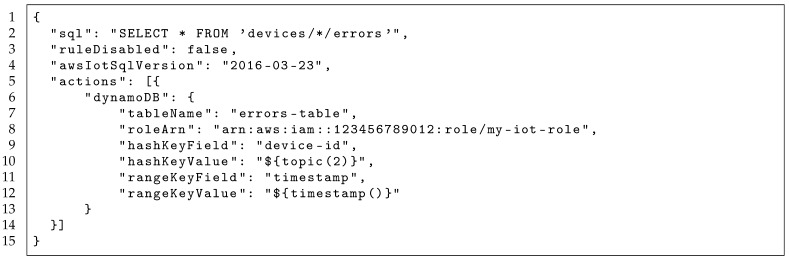
A rule to process and store error messages into DynamoDB.

This error logging system complements the default error logging on AWS (CloudWatch) and allows us to view errors affecting the daemon without physical access to the devices. Of course, this approach only works it the errors are not related to the connection itself; in that case, it is necessary to check the error logs stored on the device. Figure 4 shows the complete flow of the data and error logging functionalities of the system.

### 4.2. Proof of Concept

A pilot test was carried out with end-user devices delivered by Plan Ceibal to validate the prototype. The data collection daemon was installed in more than 2000 devices using Plan Ceibal software update tools. This test lasted one month in October–November 2017, and it was useful not only to verify the platform operation but also to collect parameters from the devices which had never been monitored before by Plan Ceibal. In effect, parameters are measured using standard Linux OS utilities running on the device, and therefore the possibilities for parameter monitoring is only limited by OS capabilities, in the case of the prototype, Ubuntu 14.04. Plan Ceibal uses different hardware platforms and operating systems; the prototype comprised two models with Intel Celeron N3160 4-core CPU @1.60Ghz and IEEE 802.11b/g wireless connectivity: (i) Clamshell laptops with 2 GB RAM and 32 GB of eMMC memory, and (ii) Positivo laptops with 2 GB RAM and 16GB SSD mSATA memory.

It is important to consider the cost of running the system over a public IoT platform, in this case, AWS. In particular, let’s consider Lambda cost analysis. Lambda charges the user based on the number of function requests and the duration, i.e., the time it takes for the code to finish executing rounded up to the nearest 100 ms. This last charge depends on the amount of RAM the user allocates to their functions. A basic free tier is offered for the first one million requests and 400,000 GB-Seconds of computing time per month. At the time the test was performed, the charge was 0.2 US Dollars per one million requests, and 0.00001667 US Dollars for every GB-Second used after that [26]. The deployed prototype did not surpass the free-tier limits while running.

Currently, the system stores collected data for later analysis, but it may be used to feed on-line analytics. To illustrate the kind of analysis that can be performed over collected data, we present two examples. The first one uses the information corresponding to the battery charge of the devices, while the second one analyzes the locations where the devices are used, distinguishing *at school* use from *at home* use.

#### 4.2.1. Battery Charge Analysis

Battery charge is an important parameter that allows analyzing relevant aspects such as battery performance, the typical users’ habits for charging the devices, and the equipment autonomy drift as time passes. Moreover, this type of analysis may be used to trigger preventive maintenance or replacement actions, in order to shorten downtimes caused by hardware failures. Figure 5a shows the battery charge empiric distribution obtained from the collected data, considering every measurement individually, and without any aggregation per device. It is worth noting that fully charged devices connected to an external power supply explain the peak in the last bin. We guess so because the reported battery charge parameter value is 100% in those cases. Collected data approximates quite well to a uniform distribution leaving aside the last bin. Next, the data is aggregated taking the median value by device. Figure 5b corresponds to the resulting distribution, which as expected is well approximated by a Gaussian distribution since the median and the average are quite similar for the collected data. Finally, Figure 5c shows a time series example for a specific device. The missing values correspond to periods when the device was off or had no connection to the network; therefore no data is reported. In this case, the discharge/charge cycles of the device can be observed looking at the temporal evolution. This information would be beneficial for device management (e.g., for selecting better battery suppliers, detecting a malfunction, among many others).

We delve into collected data to analyze the battery performance, considering only the two laptop models with the most significant amount of devices involved in the pilot (called A and B from now on). First, we processed the time series of the battery charge measurements and calculated the discharge coefficient for each device. For this purpose, a linear regression of the discharge curves was performed, which seemed an appropriate model for the observed data (cf. Figure 5c). Finally, in order to have only one discharge rate per device, we took the median value from all the estimates.

Figure 6a shows the empiric distribution for the estimated discharge rates for each laptop, considering the values for devices of type A and B. The discharging coefficient is expressed in percentage of charge per unit of time, thus it indicates the battery discharge per second. Using these values, it is possible to calculate the device estimated autonomy (i.e., the time a fully charged laptop takes to discharge its battery completely).

Considering battery performance, it is more useful to examine the devices autonomy as a metric instead of the discharging coefficient. Therefore, for the two laptop models analyzed, we compared the devices’ estimated autonomy with measurements taken in lab conditions by Plan Ceibal using their standard equipment evaluation processes. For each device, two autonomy tests are usually carried out by Plan Ceibal, one in low power usage conditions (screen with low brightness and the device running no activity) and another in high battery consumption conditions (maximum screen brightness and device playing HD video). Figure 6b,c, present the estimated autonomy empiric distribution for each device. The vertical red lines correspond to the lab measurements in low and high consumption conditions respectively. As we can see, for both laptop models, most of the devices have estimated values (calculated from the field measurements) within the range given by the minimum and maximum autonomy values measured in the lab environment.

For type A devices, 51% of them fall within the range from lab measurements, while for type B devices the percentage rises to 92%. Devices with an estimated autonomy above the maximum measured in the lab may correspond to low power usage conditions, which consume less battery than the extreme conditions used in the lab test. Furthermore, with this comparison, it is possible to identify those batteries with performance worse than expected (i.e., those below the minimum autonomy measured in the lab environment).

#### 4.2.2. Use by Location

As already mentioned, every student that assists to primary and secondary public schools obtains in property a device with wireless capabilities. They carry their laptops from home to school and back every day. It is therefore interesting to compare the use of the device along the day and to distinguish at school from at home use. This analysis is relevant, because making decisions based only on at school use may lead to the wrong conclusions. Although in many cases devices are hardly used at school, they are widely used at home. This fact has to be considered when assessing the impact of Plan Ceibal and the fulfillment of its goal of fighting the digital divide. We can indirectly measure if the device is used, and where it is used, using our tool, by considering that the device is in use if we receive data from it. It is important to notice that this may underestimate at home use since we are not able to measure if the device is off-line, while at school off-line use is not frequent since internet access is always available. We first analyze the average amount of days that each device reported data during our pilot study, and using the IP addresses and WiFi SSID information we distinguish at school use from outside school use. Figure 7a presents these data, showing that devices are used more outside the school than at school. We then refine this analysis. Figure 7b shows the average connected hours per day, while Figure 7c shows the percentage of time connected to the internet that corresponds to at school connections. This last figure clearly shows that beneficiaries use their devices at home more than at school. Finally, we analyze the evolution of this indicator. Figure 7d presents the average connection time per device and per day, showing that at school use is always below 50% and that during the weekend at school use drops to least than 10%.

These simple examples show the potential offered by the developed platform. The first one shows that it can be used for preventive maintenance, in this case, applied to batteries. It is possible to identify batteries in lousy condition, enabling, for example, automatic user alerts for battery replacement, contributing to improve user experience, as mentioned before. The second one shows that it can also be used to analyze user behavior, and with the associated analytics it can provide useful insight to decision making processes. The system administrators may develop particular data analysis based on the collected data, and, depending on the envisioned use cases, they can tune the parameters under monitoring, by merely developing scripts running on the devices, which automatically will collect data after a software update.

As mentioned in Section 4.1, the prototype implements data retrieval and storage, offering capabilities to build a business layer on top. In particular, we envision that proactive management processes based on online analytics should be the core of such a business layer, driven by Plan Ceibal needs; the aforementioned preventive battery replacement is a clear example. Other possible scenarios include (i) proactive capacity provisioning for school networks which exhibit QoS degradation signs, and (ii) laptop upgrades based on CPU, memory and storage usage, among many other possibilities.

## 5. Conclusions

One-to-one computing educational programs comprise a growing management complexity, including the interaction of ERP/CRM, Network, Device, and Learning Platforms management systems, in order to build advanced services, such as QoS guarantees to students and teachers depending on their behavior. Mobile Device Management of different terminals such as laptops, tablets, smart-phones with diverse hardware and operating systems is particularly challenging, and commercial solutions, apart from being costly, frequently do not cover every possibility. In this paper, we propose to apply IoT concepts to the MDM problem, where each mobile device acts as a sensor/actuator and the primary target is the device itself, i.e., gathering internal variables such as CPU and battery usage and temperature, on-off periods, application usage among many other possibilities. Devices can also help network management, for example collecting network traffic statistics, WiFi signal strength and coverage combined with location.

The development of IoT applications is hardly standardized, and therefore finding design patterns is of supreme importance. After surveying state of the art, we focused on a particular platform, AWS IoT. Developing an MDM module over an IoT cloud platform provides many advantages over a traditional approach. First, it is simple to develop and deploy daemons for new devices. As long as they implement the protocol, there is no need for changes in the back-end modules. Second, the new devices will be automatically added to the registry and will be able to send messages afterward. Third, since the module is in the cloud, scaling is simple and happens automatically.

The major downside of our approach is its dependency on a proprietary platform. We identify platform-independent aspects of the development, stating a clear migration path to other cloud-based (or owned) platforms, preserving application logic and data. If migration were necessary, every service should be re-implemented using the equivalent service of the target platform. However, our solution does not use any platform specific feature, especially we avoid MQTT broker advanced features such as *Thing Shadows*; the Lambda functions can be run in a dedicated machine if necessary, while the primary database could be run on any machine, and a standard SQL database or non-proprietary No-SQL database can implement the error logs database. The daemon running in the devices does not use Amazon’s MQTT library, making it easier to migrate hosts. The most difficult parts of the system to migrate would be the rules system and permissions since even though every platform has one, the implementations vary greatly.

We deployed our prototype over 2000 devices for over a month, which enabled us to test the intended functionality and gather some sample data to perform preliminary analytics. A simple analysis of battery performance found a strong correlation between the battery usage measured with the prototype, compared to the battery tests performed in laboratory conditions. Measuring real battery performance would permit to perform preventive maintenance, improving the user experience. We also analyzed the usage of the device along the day, seeking to distinguish at school from at home usage. We found that at home usage is longer on average than at school usage, and this result is relevant considering that one of the main goals of Plan Ceibal is bridging the digital divide.

This kind of analytics over the gathered data may help decision-making and planning processes, seeking for improved user experience, and consequently a higher impact of the solution. Indeed, giving the platform better proactive properties is being considered as future work, together with the careful analysis of the architecture (including the cloud platform used) needed for the commissioning phase.

Overall, the proposal delivers a fair, scalable and extensible solution to MDM in the context of one-to-one programs.

## Figures and Tables

**Figure 1 sensors-19-00070-f001:**
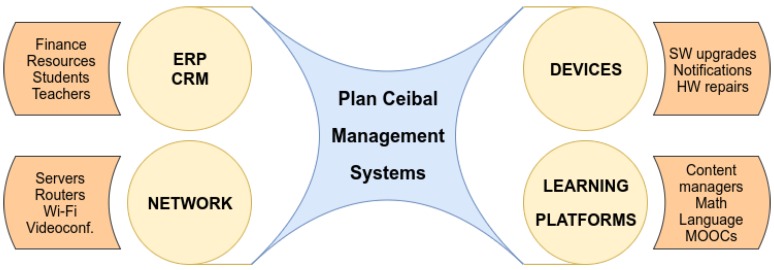
Management systems at Plan Ceibal.

**Figure 2 sensors-19-00070-f002:**
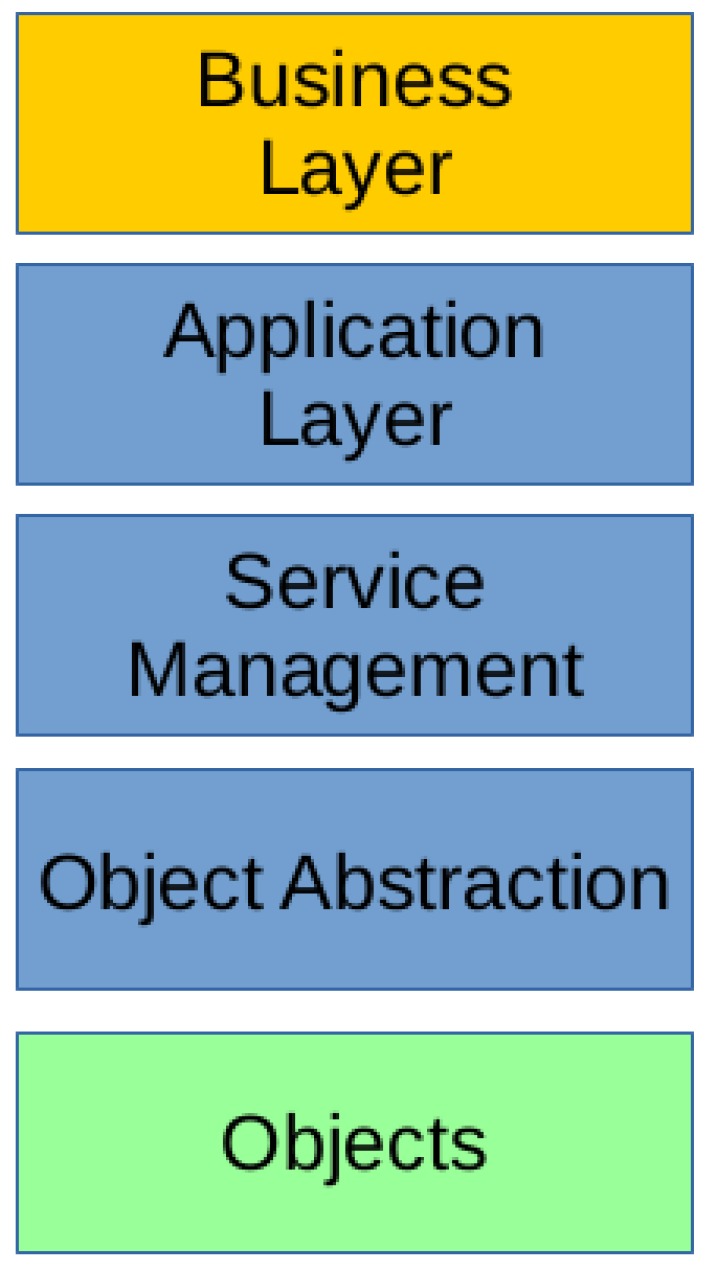
Five-layer IoT architecture (adapted from Al-Fuqaha et al., 2015 [16]).

**Figure 3 sensors-19-00070-f003:**
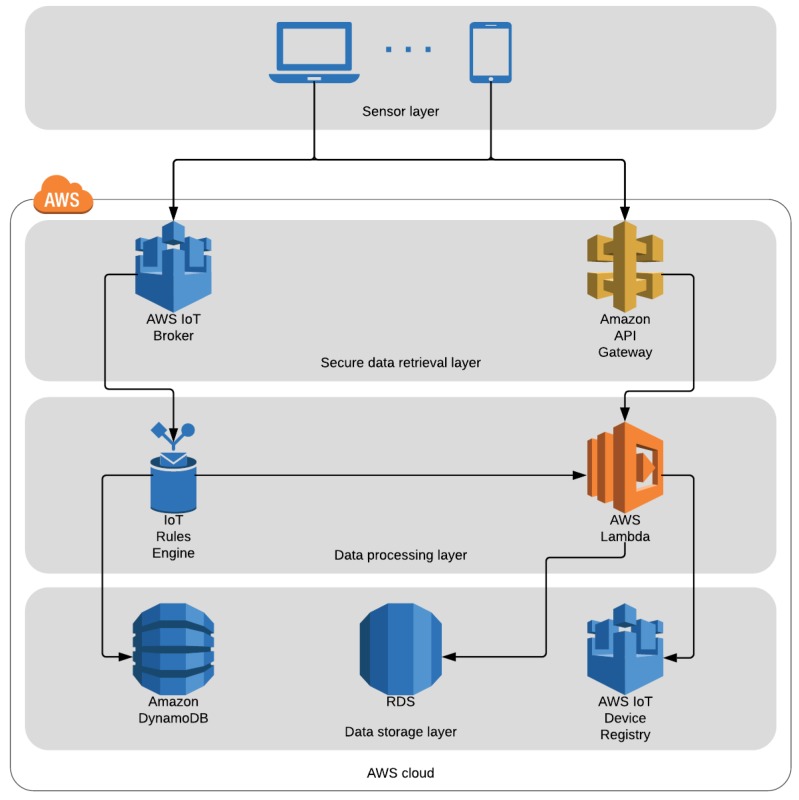
Prototype architecture.

**Figure 4 sensors-19-00070-f004:**
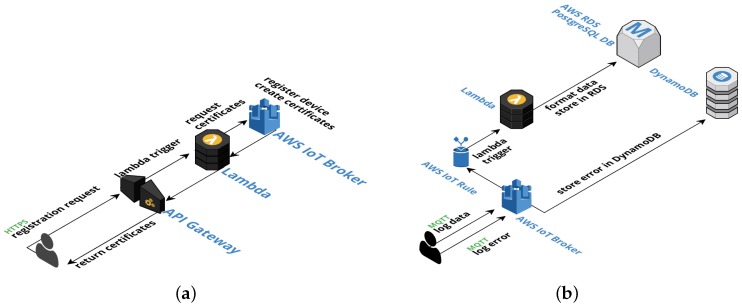
Register new device and error logging flow and architecture of the prototype. (**a**) Register new device flow and architecture. (**b**) Data and error logging flow and architecture

**Figure 5 sensors-19-00070-f005:**
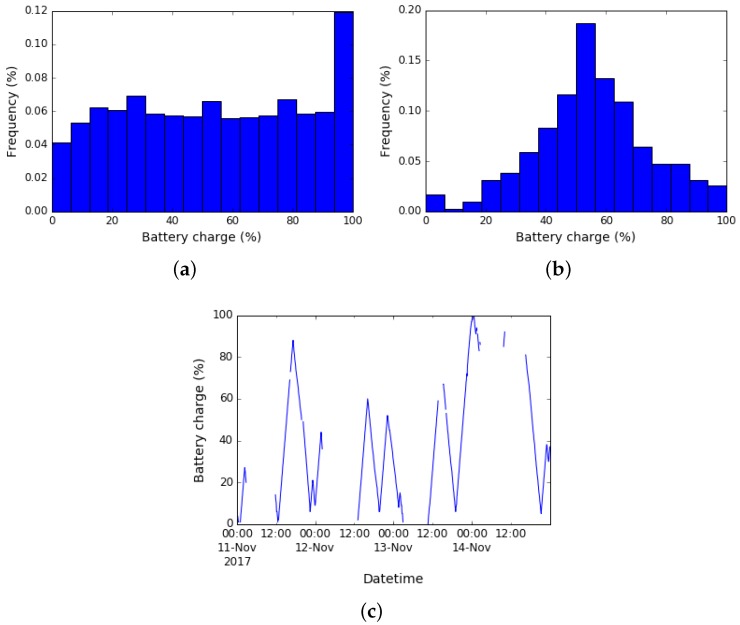
Time series example: battery charge. (**a**) Battery charge histogram. (**b**) Battery charge histogram, considering the median value per device. (**c**) Example of battery charge time series, corresponding to one particular device during four days.

**Figure 6 sensors-19-00070-f006:**
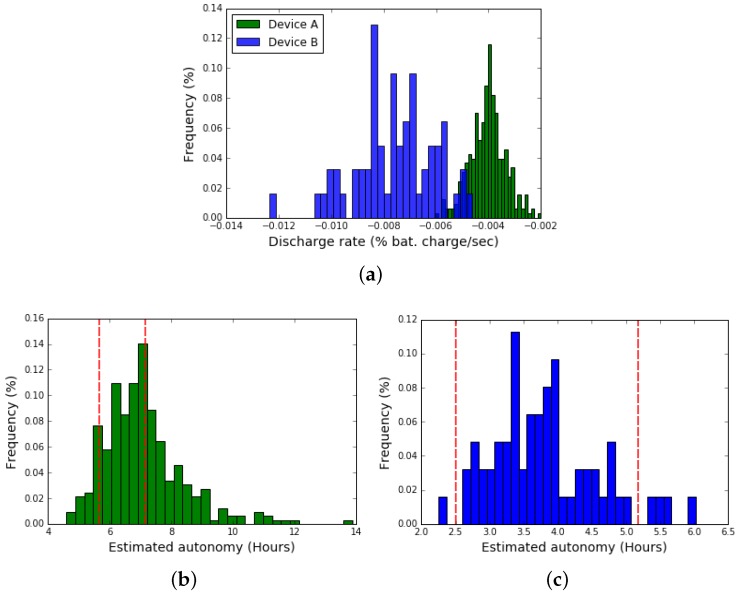
Discharge rate distribution and estimated autonomy analysis. (**a**) Discharge rate distribution for devices of type A and B. (**b**) Estimated autonomy for devices of type A. (**c**) Estimated autonomy for devices of type B.

**Figure 7 sensors-19-00070-f007:**
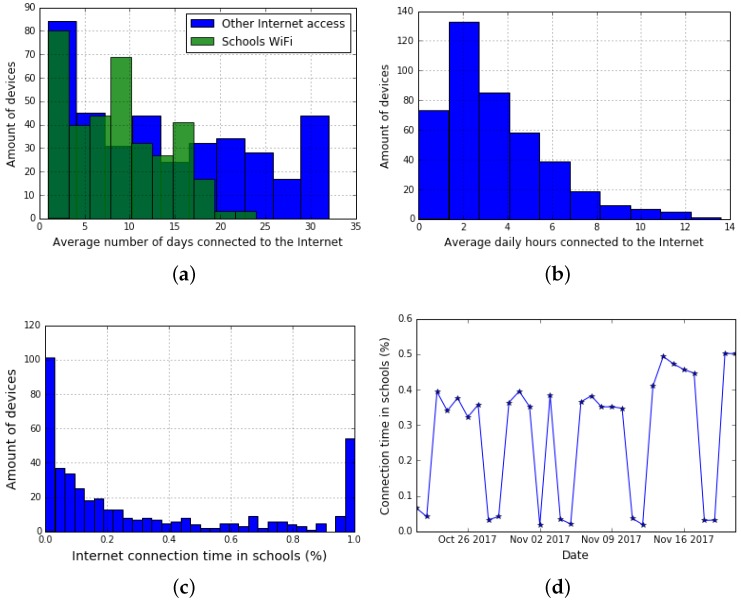
Use by location example. (**a**) Average number of connected days. (**b**) Average connected hours per day. (**c**) Percentage of at school connection time. (**d**) Percentage of at school connection time by day.

**Table 1 sensors-19-00070-t001:** Logged data from devices.

Attribute	Description
mac_addr	Device’s MAC address.
serial_number	Device’s serial number.
ip_addr	Device’s public IP address.
timestamp	Added timestamp.
ap_mac_addr	MAC address of the access point the device is connected to.
frequency	Frequency of the network the device is connected to.
rssi	Relative received signal strength of the network the device is connected to.
tx_packets_quantity	Information about the packages transmitted by the interface.
tx_packets_overruns	
tx_packets_carrier	
tx_packets_errors	
tx_packets_dropped	
tx_excessive_retries	
rx_packets_quantity	Information about the packages received by the interface.
rx_packets_overruns	
rx_packets_frame	
rx_packets_errors	
rx_packets_dropped	
rx_bytes	
charging	Indicates if the device is charging or not.
battery_temp	Battery temperature.
battery_power	Battery charge level.
uptime	Time the device has been powered-on.
boot_time	Last boot time.
load_avg_5_min	Average load of the CPU in the last five minutes.
total_memory_kb	Total RAM capacity.
free_memory_kb	Free RAM.
total_swap_memory_kb	*Swap* memory size.
free_swap_memory_kb	Free *swap* memory.
cached_memory_kb	Page cache size.
buffers_memory_kb	I/O *buffers* size.
root_dir_total_disk_space_kb	Total space on the device’s /root directory.
root_dir_free_disk_space_kb	Free space on the device’s /root directory.
home_dir_total_disk_space_kb	Total space on the device’s /home directory.
home_dir_free_disk_space_kb	Free space on the device’s /home directory.

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
