# Peer review of "Managing Devices of a One-to-One Computing Educational Program Using an IoT Infrastructure"

_sensors, 2018, doi:10.3390/s19010070_

Round 1
Reviewer 1 Report
The presented topic is interesting and the availability of a plethora of heterogeneous devices to be monitored and managed is challenging. However, in my opinion authors should greatly improve the technical level of the submitted manuscript. In particular, authors should provide many more details about their implemented solution, while (eventually) shortening the general description of adopted models and technologies (see Section 2 and Section 3.1).
Just to provide some relevant examples, the technical quality of the manuscript would be improved by:
- detailing the protype architecture (e.g., adding a figure in Section 4.1),
- explaining why the proposed solution can be regraded as server-less (better explaining the meaning of this term and the implications for the project, also in terms of costs due to IoT AWS pricing),
- providing clear examples of rules enforced on AWS and of Lambda functions processing the data,
- better describing developed API for the API Gateway, how they hide specific characteristics of AWS and where they run (in my opinion, the API Gateway can be considered as a server for the client, thus it seems that the solution is not completely server-less),
- specifying when the test has been carried on and which are the hardware/software capabilities of monitored laptops,
- clearly defining a metric for the discharging coefficient,
- detailing Linux commands exploited to monitor the state of devices,
Moreover, I would also appreciate the availability of additional quantitative results. In particular, it would be very interesting to show how monitored information are (or at least can be) exploited to proactively manage devices. In fact, while the abstract cites the possibility of performing preventive and proactive device management, the rest of the paper does not clearly how it is actually performed, leaving the impression that the abstract overpromises in relation to the actual project.
Minor comments:
- please, be sure to have the right of providing Figure 2 and Figure 3 in relation to the copyright
- Section 3.1: “Kaa and Fiware” in place of “Kaa y Fiware”
Author Response
The presented topic is interesting and the availability of a plethora of heterogeneous devices to be monitored and managed is challenging. However, in my opinion authors should greatly improve the technical level of the submitted manuscript. In particular, authors should provide many more details about their implemented solution, while (eventually) shortening the general description of adopted models and technologies (see Section 2 and Section 3.1).
Just to provide some relevant examples, the technical quality of the manuscript would be improved by:
- detailing the protype architecture (e.g., adding a figure in Section 4.1),
- explaining why the proposed solution can be regraded as server-less (better explaining the meaning of this term and the implications for the project, also in terms of costs due to IoT AWS pricing),
- providing clear examples of rules enforced on AWS and of Lambda functions processing the data,
- better describing developed API for the API Gateway, how they hide specific characteristics of AWS and where they run (in my opinion, the API Gateway can be considered as a server for the client, thus it seems that the solution is not completely server-less),
We re-wrote almost completely Section 4 "Description of our proposal" to take into account the reviewer comments. In particular present Figure 3: Prototype architecture, showing a four-layer implementation, namely Sensor, Sensor data retrieval, Data processing, and Data storage layers; we also loosly mapped this architecture to the generic five-layer architecture presented in Section 3.1.
We explained the term server-less, and how this apply to our prototype in Section 4.1.2 "AWS layers", providing examples of rules enforcement in lines 381 to 392 of the manuscript, including a real example in Listing 1: A rule to process and store error messages into DynamoDB.
We provided a cost analysis of the usage of Lambda functions in Section 4.2 "Proof of concept", in lines 406 to 412 of the manuscript.
- specifying when the test has been carried on and which are the hardware/software capabilities of monitored laptops,
- clearly defining a metric for the discharging coefficient,
- detailing Linux commands exploited to monitor the state of devices,
We provided a better description of the test characteristics in Section 4.2 "Proof of concept"; in particular we stated the test period (October-November 2017), and the hardware and software capabilities of the monitored laptops.
We provided a short list of the linux utilities used for gathering data in lines 340 to 341 of the manuscript, also including a detailed list of gathered parameters in Table 1. Logged data from devices.
Regarding battery discharging coefficient, the topic is detailed in lines 442 to 450 of the manuscript.
Moreover, I would also appreciate the availability of additional quantitative results. In particular, it would be very interesting to show how monitored information are (or at least can be) exploited to proactively manage devices.
In fact, while the abstract cites the possibility of performing preventive and proactive device management, the rest of the paper does not clearly how it is actually performed, leaving the impression that the abstract overpromises in relation to the actual project.
We re-wrote the abstract, clearly stating that the prototype is limited and the proactive management capabilities could be built on top of the data collection infrastructure. We also mentioned the topic in Section 4.1 "Prototype architecture", regarding the business layer of the reference architecture, in lines 332 to 335. Finally, we tackle the issue at the end of Section 4.2 "Proof of concept", providing some scenarios in lines 493 to 499 of the manuscript.
Minor comments:
- please, be sure to have the right of providing Figure 2 and Figure 3 in relation to the copyright
- Section 3.1: “Kaa and Fiware” in place of “Kaa y Fiware”
Both issues have been solved. We retired the mentioned figures, and provide a description in one case, and our own figure, adapted from Al-Fuqaha et al., in the other.
p { margin-bottom: 0.1in; line-height: 115%; }p { margin-bottom: 0.1in; line-height: 115%; }
Reviewer 2 Report
You should avoid taking figures from other papers even if you cite them (figure 2, figures 3)
For IoT, cloud and Big Data you could take a look at: https://www.mdpi.com/1424-8220/18/4/1181
The data set seems interesting, but as it is used, the article provides limited contributions. You should try and find similar project that collect similar data and insist on what your project does differently. Measuring wireless networks from the end-user’s devices perspective is not exactly new.
Do you really need AWS for the back-end infrastructure?
Author Response
You should avoid taking figures from other papers even if you cite them
(figure 2, figures 3)
We retired the mentioned figures, and provide a description in one case, and our own figure, adapted from Al-Fuqaha et al., in the other.
For IoT, cloud and Big Data you could take a look at: https://www.mdpi.com
/1424-8220/18/4/1181
We included this reference together with other references in the field already considered in the paper.
The data set seems interesting, but as it is used, the article provides limited contributions. You should try and find similar project that collect similar data and insist on what your project does differently. Measuring wireless networks from the end-user’s devices perspective is not exactly new.
Regarding originality of the contribution, we insist that the specific field of Mobile Device Management (MDM) using IoT concepts is relevant, and building solutions architecturaly sound for open source systems is a valuable contribution both academic and for practiotioners, in this case, Plan Ceibal. Monitoring wireless parameters on users' devices is not new, but this aspect is an add-on of our main goal, which is measuring internal variables of the devices for, among other uses, repair and replacement management.
Do you really need AWS for the back-end infrastructure?
We comment this issue throughout the paper, stating that for the prototype we prioritized ease of development. We mention the topic in the abstract, and provide a comprehensive summary in the Conclusion section, in lines 520 to 530. It is also considered in lines 539 to 543 of the manuscript.
p { margin-bottom: 0.1in; line-height: 115%; }
Round 2
Reviewer 1 Report
Now the paper is worth for publication.
Author Response
Now the paper is worth for publication.
Thanks for your comment.
Regarding the point "English language and style are fine/minor spell check required ", we perform a thoroughly review of grammar and spelling, marked in green in the manuscript.
Reviewer 2 Report
Progress has been made by the authors in presenting their research.
Maybe some of the references could have been better discussed in the context of the paper and not just mentioned.
Line 330: you are missing the pointer to the reference.
Regarding the serverless architecture (lines 346-355): coming from a databases background, I usually understand by a serverless DB, an embedded solution like SQLite or H2 that has a small footprint and doesn’t require separate server processes. Now, I know that this term has been taken by the Cloud service providers with a different meaning like the one you mention. Maybe you could discuss both meanings of serverless given the fact that both are relevant in the context of your paper.
Author Response
Progress has been made by the authors in presenting their research.
Thanks for your comment.
Maybe some of the references could have been better discussed in the context of the paper and not just mentioned.
We appretiate your comment. Our intention was to point out the existence of open source, locally deployed infrastructure; since that is not our ficus, we prefered not to go deeper n these references.
Line 330: you are missing the pointer to the reference.
Fixed (now is line 328)
Regarding the serverless architecture (lines 346-355): coming from a databases background, I usually understand by a serverless DB, an embedded solution like SQLite or H2 that has a small footprint and doesn’t require separate server processes. Now, I know that this term has been taken by the Cloud service providers with a different meaning like the one you mention. Maybe you could discuss both meanings of serverless given the fact that both are relevant in the context of your paper.
We took this comment into account a re-wrote this paragraph, stating the different meanings of the word serverless. The new paragraph is marked in blue (lines 347-358).
Regarding the point "Moderate English changes required", we perform a thoroughly review of grammar and spelling, marked in green in the manuscript.
p { margin-bottom: 0.1in; line-height: 115%; }